# Single-nucleus atlas of the *Artemia* female reproductive system suggests germline repression of the Z chromosome

**Marwan Elkrewi** [ID]*, **Beatriz Vicoso** [ID]*

Institute of Science and Technology Austria (ISTA), Klosterneuburg, Austria

* marwanelkrewi@gmail.com (ME); bvicoso@ist.ac.at (BV)

## Abstract

Our understanding of the molecular pathways that regulate oogenesis and define cellular identity in the Arthropod female reproductive system and the extent of their conservation is currently very limited. This is due to the focus on model systems, including *Drosophila* and *Daphnia*, which do not reflect the observed diversity of morphologies, reproductive modes, and sex chromosome systems. We use single-nucleus RNA and ATAC sequencing to produce a comprehensive single nucleus atlas of the adult *Artemia franciscana* female reproductive system. We map our data to the Fly Cell Atlas single-nucleus dataset of the *Drosophila melanogaster* ovary, shedding light on the conserved regulatory programs between the two distantly related Arthropod species. We identify the major cell types known to be present in the *Artemia* ovary, including germ cells, follicle cells, and ovarian muscle cells. Additionally, we use the germ cells to explore gene regulation and expression of the Z chromosome during meiosis, highlighting its unique regulatory dynamics and allowing us to explore the presence of meiotic sex chromosome silencing in this group.

**Data Availability Statement:** The scripts used in the analysis can be accessed on the GitHub page: https://github.com/Melkrewi/Artemia-snRNAseq-Project. The raw data are available on the NCBI short read archive (BioProject number

## Author summary

Oogenesis is a highly complex process involving multiple cell-types and an extremely well orchestrated program that unfolds in the female reproductive system. Despite the large diversity of Arthropod reproductive modes and sex determination systems, our current understanding of oogenesis is limited to a few model species. This makes it difficult to study and formulate hypotheses about the evolutionary history, constraints, and importance of the individual elements of this process. To fill this gap, we used single-nucleus expression and chromatin-accessibility data to produce a single-nucleus atlas of the *Artemia franciscana* female reproductive system. By comparing our dataset to the published *Drosophila* single-nucleus data (over 400 million years of divergence), we were able to highlight the substantial conservation of several of the molecular pathways of oogenesis and meiosis. We found evidence of global transcriptional quiescence and chromatin condensation in late germ cells, highlighting the conserved role of this repressive stage in arthropod oogenesis. Additionally, we explored the expression patterns of the ZW sex chromosomes during oogenesis. Our data shows that the Z-chromosome is consistently

PRJNA1128544). The Seurat objects produced in the analysis, the loom files generated by Velocyto, the reciprocal best hits used for the SAMap analysis, and the module tables and their GO enrichment results are available for download on ISTA Research Explorer (ISTA REx) using the following link: https://doi.org/10.15479/AT: ISTA:17362. The single nucleus atlas can be viewed with this link on the UCSC Cell Browser: https://brine-shrimp-repro.cells.ucsc.edu.

**Funding:** This research was funded by the Austrian science fund (FWF), as part of the SFB Meiosis consortium (https://sfbmeiosis.org/, grant ID FWF SFB F88-10) to BV. The funders had no role in study design, data collection and analysis, decision to publish, or preparation of the manuscript.

**Competing interests:** The authors have declared that no competing interests exist.

downregulated in germline cells. While this is partly driven by a lack of dosage compensation in the germline, a subset of cells show stronger repression of the Z chromosome.

## Introduction

Most animals have evolved sexually dimorphic mechanisms and tissues dedicated to the production of haploid gametes through meiosis (gametogenesis). Males produce motile nuclei (sperm) through spermatogenesis, which takes place in the testes. Females produce oocytes that contain a haploid nucleus along with the cytoplasmic molecules needed to initiate and facilitate embryonic development through oogenesis, which takes place in the ovaries [1]. While many aspects of oogenesis and female meiosis are highly conserved between distant species, there is considerable diversity in many others, including the presence/absence of nurse cells, meiotic chromosome pairing strategies, recombination rates, timing and duration of meiotic arrests, and sex chromosome specific regulation [1–3]. Why such a fundamental and ancient mechanism exhibits so much variation is still unclear. Studying species with diverse body plans, reproductive modes, and sex chromosome systems at the genetic and molecular levels will help elucidate the developmental constraints and selective pressures that shaped the evolution of the conserved, convergent and divergent features of oogenesis.

As the hallmark of oogenesis, meiosis is very tightly regulated and many mechanisms have evolved to ensure the faithful transmission of genetic information to the offspring through the proper pairing and segregation of homologous chromosomes [4]. In addition to ensuring the fidelity of the transmitted genome copy and building the maternal reserves to kick-start the embryo's journey, oogenesis involves extensive reprogramming of the epigenetic landscape to ensure a successful oocyte-to-embryo transition upon fertilization [5]. To accomplish those feats, oocytes actively navigate previtellogenesis and the majority of prophase I before they arrest and become transcriptionally quiescent. The prophase I arrest is thought to be essential for oocyte growth and differentiation, and is conserved across metazoans, but with highly variable durations [6]. In many arthropods, the lack of transcription in the oocyte is buffered by the activity of nurse cells that remain connected to the oocyte through cytoplasmic bridges [7]. Sister cyst cells have been shown to play a similar role in mice, suggesting they might also be important for mammalian oocyte differentiation [8]. In addition to this buffering, work in *Drosophila* suggests that some oocyte specific transcription, regulated through epigenetic programming in early oogenesis, occurs before the resumption of meiosis [9]. This, along with the findings that oocyte chromatin enrichments of H4K16ac and H3K27me3 are maintained in the oocyte-to-embryo transitions in *Drosophila* and mammals, highlights the importance of the epigenetic regulation in early oogenesis [10–12]. Somatic-germ cell signaling is also known to play an important role in oogenesis, where in *Drosophila* and mammals, signaling, often hormonal, by the surrounding follicle cells plays an important role in triggering oocyte maturation [13,14]. Despite our understanding of some of the pathways involved, many questions still surround the intrinsic and extrinsic signaling involved in oocyte differentiation, maturation and the resumption of meiosis after the long arrest, including their relative contribution, how they evolved in the first place, and how conserved they are across metazoans.

The presence of differentiated/heteromorphic sex chromosomes, such as the X and Y pair of mammals, introduces two challenges: First, X-linked genes are found in different copy numbers between males and females, which can cause imbalances in expression; second, sequence similarity is usually required for successful synapsis and accurate segregation of homologous chromosomes, but is lacking over most of the length of the XY pair. Dosage compensation

mechanisms have evolved to tackle the first problem, with some species, such as *Drosophila*, upregulating the expression of the single X in males, and others, such as mammals, inactivating one of the X chromosomes in females [15]. This sex chromosome-specific epigenetic regulation seems to disappear/reverse in the germline cells of *Drosophila* males [16,17] and undergo extensive reconfiguration in mammalian females, with a period of hyper-transcription before reaching dosage balance (X:A ~ 1) [18,19]. It is not clear whether this absence of dosage compensation in the germline is linked to the epigenetic reprogramming that takes place during gametogenesis, and whether it is the rule or the exception in Arthropods.

The second problem is germline specific, as the sex chromosomes fail to synapse across some or most of their length during prophase I [20]. Asynapsis of other chromosomes triggers germ cell arrest as a defense mechanism against genome instability and aneuploidy [21,22], and mechanisms must be in place to ensure that meiotic cells pass this checkpoint in the presence of unpaired sex chromosomes. This has been extensively studied in mammalian spermatogenesis, where the non-recombining regions of the XY chromosomes fail to pair and remain unsynapsed during meiotic prophase I [23]. This leads to the accumulation of repressive chromatin marks and silencing along the two sex chromosomes. The XY chromatin condenses and forms what is called the sex body (XY body), which is inaccessible to the transcription machinery [24,25], leading to the complete silencing of both chromosomes. This process is termed meiotic sex chromosome inactivation (MSCI), and it has been described in marsupials, eutherians, nematodes, beetles, chickens, and fruit flies, with the latter two cases being disputed afterwards [26–29,22,30]. In the case of fruit flies, a recent study showed that the X chromosome of *Drosophila melanogaster* is not enriched for silencing marks in spermatocytes, suggesting the absence of MSCI [31]. Another study used single-cell RNA-seq in *Drosophila miranda*, which has the ancestral *Drosophila* X (Muller element A), along with two younger X-linked chromosome arms (Muller AD and Muller C), found that all three X chromosomes have expression patterns consistent with a lack of dosage compensation in late spermatocytes and spermatids [32]. The presence of MSCI in some but not all organisms with differentiated XY chromosomes raises the question of what drives it to evolve in the first place, and what alternative mechanisms may be in place in species lacking it.

Currently, the understanding of the sex chromosome specific regulation and the interplay with the tight constraints of gametogenesis is biased towards model species with XY chromosomes. Species with ZW chromosomes (females are ZW, males ZZ) provide an interesting counterpart, as the dosage imbalance and the pairing issues will occur in the female rather than male and during oogenesis rather than spermatogenesis. The status of MSCI in ZW systems is unclear: as after a study reported its presence in the ZW system of chicken during oogenesis [26], another study came to the opposite conclusion [27]. A study of MSCI in two Lepidoptera species reported that the Z is euchromatic and transcriptionally active during meiosis [33]. However, none of these studies quantified transcriptional output directly, and partial reductions in sex chromosome expression may have been missed. Additionally, as dosage compensation mechanisms evolved to mask the deleterious effects of having a single copy of dosage-sensitive genes, imbalances in the germline of heterogametic females should hypothetically have detrimental effects on oogenesis and early embryogenesis. Due to the limited number of studies on ZW systems, whether dosage compensation is present in the germline cells and whether Z-chromosomes are inactivated during oogenesis are still open questions.

Here, we address these questions using *Artemia* brine shrimp, an aquatic arthropod from the Branchiopoda class with a pair of differentiated ZW sex chromosomes [34]. Arthropods have two major types of ovaries: panoistic, where all the germline cells differentiate into oocytes, and meroistic, where only one cell becomes an oocyte and the rest of the germ cells differentiate into nurse cells. Although the majority of crustaceans have panoistic ovaries,

meroistic ovaries are typical for Branchiopoda [35]. This facilitates drawing parallels between the *Artemia* reproductive system and that of *Drosophila melanogaster*, the most studied insect system in terms of molecular, developmental and morphological data.

In *Artemia* females, oogenesis starts in the two tubular-like ovaries, where the germ cells differentiate into oocytes and up to 70 nurse cells, and where most of the previtellogenesis and vitellogenesis take place [35]. *Artemia* nurse cells and oocytes do not show any differences in their morphology until the end of previtellogenesis, where the nurse cells reportedly become polyploid (to increase ribosomal RNA content), and unlike oocytes, do not undergo vitellogenesis and do not produce yolk protein [36]. Similar to *Drosophila*, nurse cells remain connected to the oocyte through cytoplasmic bridges and continue supporting it until the end of vitellogenesis, where they are phagocytosed by follicle cells. The oocytes progress through prophase I as they grow in the ovary and move towards the oviduct, where they stay temporarily. After that, the eggs move to the ovisac and stay there in arrested metaphase I until fertilization [36,37].

Although an analysis in the water flea *Daphnia* [38], the closest model organism to *Artemia*, suggests the sequence conservation of many meiosis genes between insects and crustaceans, their conserved role in crustacean oogenesis and meiosis, and the transcriptional diversity/heterogeneity of cell types in the female reproductive system, have yet to be studied. Here, we create a single nucleus RNA sequencing (snRNA-seq) atlas of the *Artemia* ovary, and we identify different somatic and germline cell types, allowing us to perform a detailed comparison with the well-characterized *Drosophila melanogaster* [39]. We further combine RNA-seq and chromatin accessibility (ATAC-seq) data obtained from the same nuclei to investigate the transcriptional and epigenetic dynamics of germ cells during oogenesis. Finally, we characterize the dynamic expression of Z-linked genes in oogenesis, to test whether dosage compensation and sex chromosome inactivation occurs in the germline of the independent ZW system of *Artemia*, enhancing our understanding of sex chromosome regulation during meiosis.

## Results

### 1. snRNA-seq identifies unique cell clusters that share conserved expression programs with *Drosophila*

To resolve the cellular heterogeneity in the *Artemia* female reproductive system and explore the unique regulatory programs and chromatin accessibility in the different cell types, we performed 10x single nucleus RNA sequencing experiments on two biological replicates of pooled ovaries from females kept with males (and therefore putatively mated), and two replicates of 10x Multiome ATAC+Gene expression experiments on pooled ovaries from unmated females (isolated in individual vials after birth to ensure that germ cells would not progress past metaphase I). After preliminary quality checks, and removal of ambient RNA and contamination with cells, we integrated the gene expression data from the four replicates and used dimensionality reduction algorithms to cluster the 20,109 remaining nuclei into 7 clusters (Fig 1A), one of which seems to be specific to the individuals which had access to males (S1 Fig, S1 Table). The clustering resolution (0.05) was chosen based on the specificity-based resolution selection criterion approach (S2 Fig) [40]. Heatmaps of the top 10 markers for each cluster identified using Seurat [41] functions (S3 and S4 Figs) suggest that these correspond to functionally differentiated cell types. In order to annotate these distinct cell types, we mapped our clusters to the *Drosophila* ovary dataset from the Fly Cell Atlas [42] using SAMap [43], and filtered for an alignment threshold above 0.2. All the different clusters in *Artemia* map to *Drosophila* clusters (Fig 1B), supporting a high level of conservation of the molecular pathways that define cellular identity in the ovary. Two of our clusters map to germline cells in the *Drosophila* dataset: one to the cells from the germarium region, which we labeled as Germ cells A, and the other to all

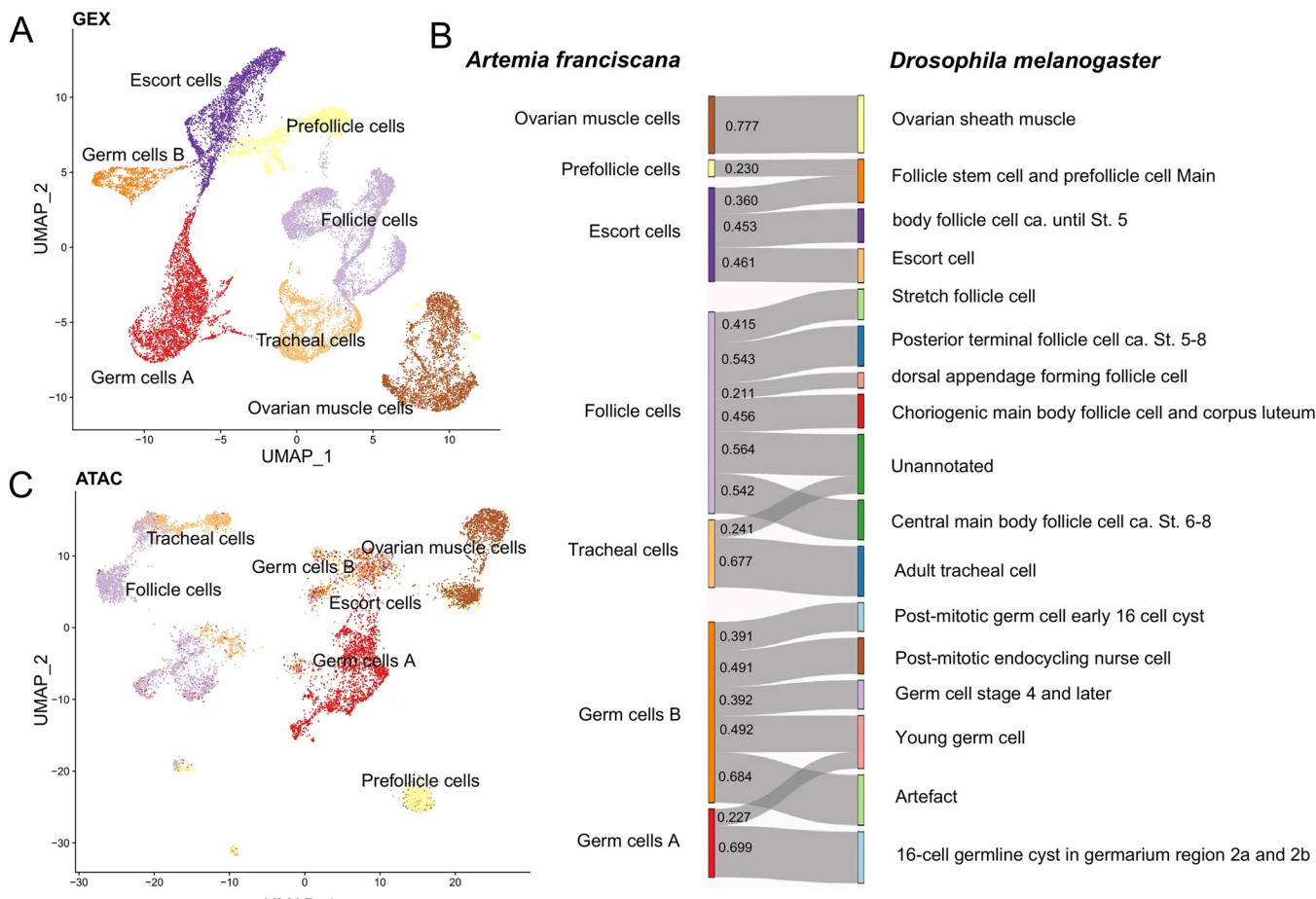

**Fig 1. Conserved cell types between *Artemia* and *Drosophila*.** A) Gene expression UMAP. B) Sankey plot showing the mapping between the *Artemia* and *Drosophila* clusters, the number corresponds to the alignment scores between the clusters. C) UMAP based on the ATAC-seq data colored based on the gene expression cell-cluster assignments.

the later stages of germline cell differentiation, which we refer to as Germ cells B. The expression correlation matrix and its corresponding dendrogram (S5 Fig) show that the clusters mapping to Escort cells and Prefollicle cells are nested with the germ cells, suggesting that those labels might not accurately describe the role of those cell clusters in *Artemia*. To account for this, we defined three major groups (based on the correlation and UMAP distance): group 1 includes Germ cells A, Germ cells B, Escort cells, and Prefollicle cells, group 2 includes Follicle and Tracheal cells, and group 3 includes Ovarian muscle cells.

We also assessed whether the ATAC-seq data contains enough information to disentangle the different cell types in our samples, and whether they would correspond to the cell types inferred from the expression data. We called peaks using MACS2 [44] in each cell type and clustered the nuclei using the peaks. Similar clusters were recovered when using the ATAC data as from the RNA-seq (Figs 1C and S6), providing further support for their validity.

## 2. Germ cells express conserved germline markers and are enriched for meiosis associated genes

To validate the early and later germ cell assignments, we checked the expression of known conserved markers for early (Orb, Fig 2A) and for late (Vas, Fig 2B) germline. Germ cells A

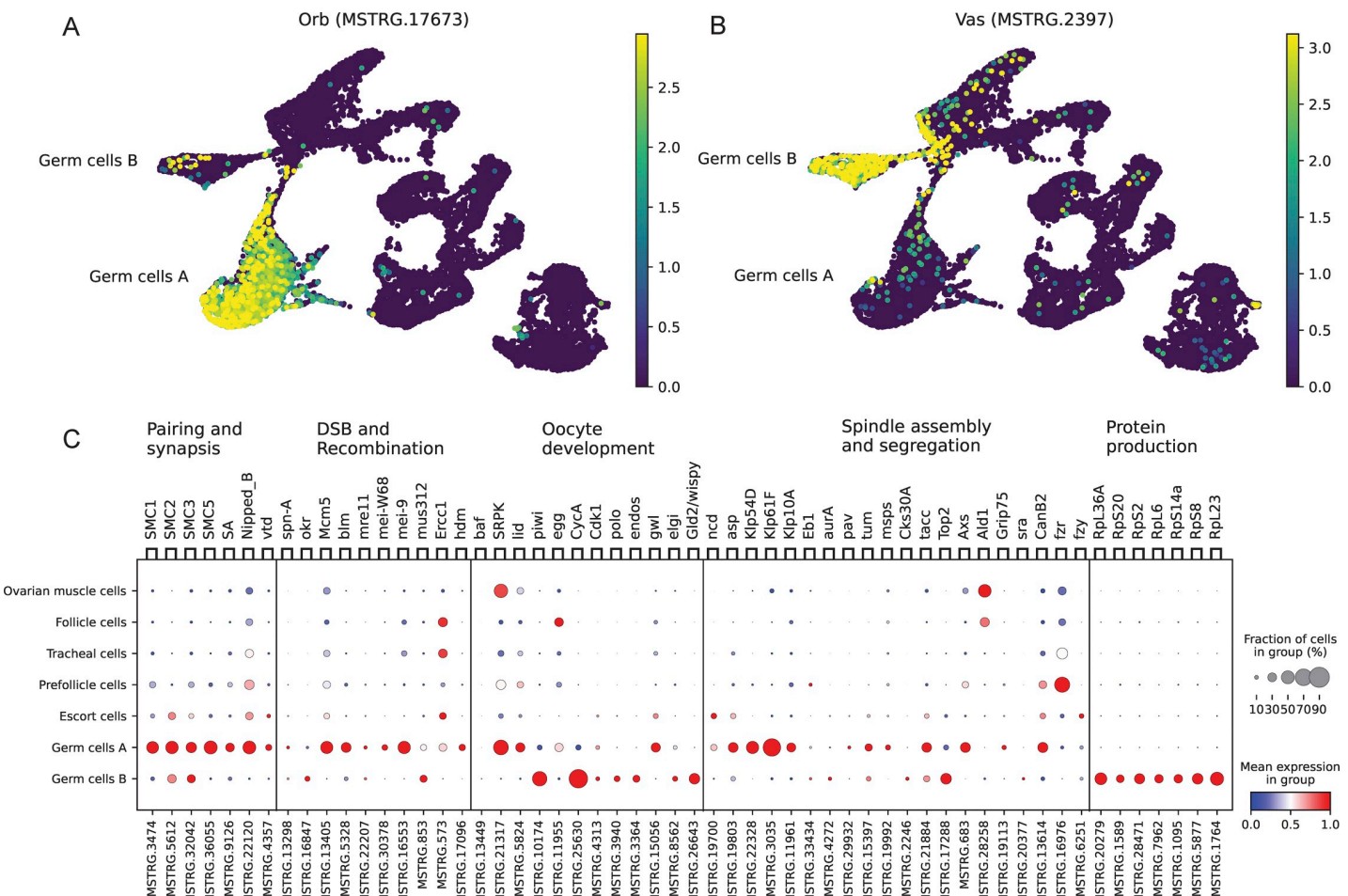

**Fig 2. Expression of *Drosophila* germ cell and meiotic markers in *Artemia*.** A) Expression of the early germline marker Orb B) Expression of the late germline marker Vasa C) The expression of *Artemia* orthologs of genes involved in the different stages of meiosis in *Drosophila*, along with the expression of genes involved in protein production.

show high expression of Orb and Germ cells B show a high expression of Vasa. This is consistent with Germ cells A being an earlier time point in oogenesis than Germ cells B, as Orb is expressed in the region 2 of the germarium in *Drosophila*, very early in oogenesis, and Vasa has been shown to be expressed in the last stages of oogenesis in *Artemia* [45–47]. In order to check for the presence of meiotic cells, and to pinpoint the meiotic stages captured in our samples, we also explored the expression of the *Artemia* orthologs of genes known to be involved in the regulation of meiosis in *Drosophila* [48] in each of the clusters (Fig 2C). None of the putative somatic clusters were systematically enriched for any category of meiotic genes. The germ cell A cluster was enriched for *Drosophila* early prophase I genes (pairing and synapsis and double-strand breaks and recombination). The germ cell B cluster was enriched for oocyte maturation (which marks the release from prophase I arrest in *Drosophila*) and germinal vesicle breakdown genes [6], suggesting that this cluster includes late prophase cells. Surprisingly given that spindle formation only occurs in late meiosis, spindle assembly genes were expressed in the early germline, but a similar pattern has been also observed in *Drosophila* [49]. It is also important to note that if the nuclear envelope disappears at the end of prophase I [36], the transcriptomes from the stages between the breakdown and reassembly of the

nuclear envelope (Telophase I) cannot theoretically be captured with single nucleus sequencing, which likely explains the predominance of prophase cells in our dataset. We also checked the expression of ribosomal proteins, as those were found to be highly expressed in the late stage germline cells of the *Drosophila* ovary single cell Atlas [47]. The Germ cells B cluster has a clear enrichment of those genes, consistent with their assignment as late stage germline cells (Fig 2C).

To further explore what pathways may be acting specifically in the germline, we used the hdWGCNA package [50] to perform co-expression network analysis and identify modules (clusters of co-expressed genes) expressed in the different cell types. We constructed networks for the whole dataset and quantified the expression of the modules in the different cell types. The analysis resulted in 14 non-overlapping modules (S7 Fig). We performed differential module eigengene (DME) analysis comparing germline (germ cells A and germ cells B) to all the other clusters and identified 5 modules upregulated in the germline clusters (Fig 3A). Of the 5 upregulated modules in germ cells, 3 had significant PPI enrichment (Figs 3 and S8, S9 and S10). Fig 3 shows modules 7 (269 genes) and 9 (250 genes), which are enriched for chromatin

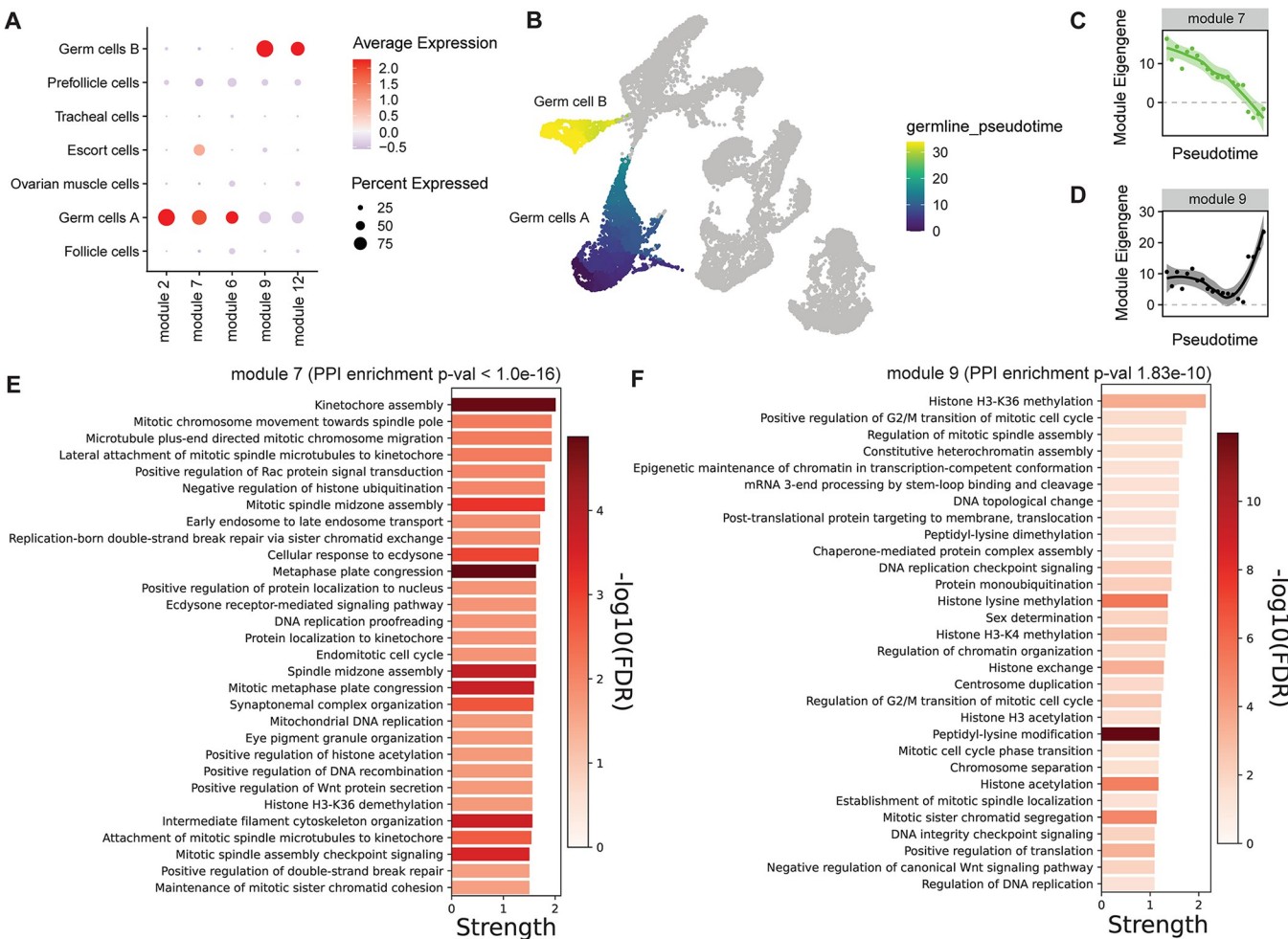

**Fig 3. Gene regulatory networks in the germline cells.** A) Dot plot depicting the expression of the modules upregulated in germ cells. B) pseudotime trajectory overlaid on the UMAP. C) Module 7 expression dynamics across the germline pseudotime. D) Module 9 expression dynamics across the germline pseudotime. E) Biological process GO enrichment in module 7. F) Biological process GO enrichment in module 9. Strength (retrieved from StringDB) is estimated as the log10(number of observed proteins with a term/ number of expected proteins with the term in a random network of the same size).

regulation/cell division related terms. Module 7 is highly expressed in Germ cells A and its expression declines across the germline pseudotime (Fig 3B and 3C). Module 9 is expressed in both germ cell clusters (Fig 3D), but peaks in expression in germ cells B. It includes many terms related to histone modifications (methylation and acetylation), further supporting the idea that extensive chromatin remodeling takes place during oogenesis. These modules further support the mitotic/meiotic activity in those clusters and provide novel candidate genes and biological pathways involved in crustacean oogenesis and meiosis.

## 3. Transcription is repressed in late germ cells

During meiotic prophase, the chromatin of *Drosophila* oocytes condenses, and this condensation is accompanied by transcriptional repression [48,51]. We explored potential signatures of such changes in our gene expression and ATAC data (replicates 3 and 4). The number of ATAC-seq counts in peaks in the Germ cells B cluster is very low compared to all the other clusters (Fig 4B, number of fragments per cluster is shown in S11 Fig), consistent with the chromosomes being highly condensed during late prophase/metaphase I. The nuclei assigned as Escort cells also show low RNA counts, likely reflecting the fact that they are misassigned Germ cells A and B (as this cluster is largely missing from our multiomics dataset). To further characterize the transcriptional activity of the different clusters, we estimated the percentage of spliced and unspliced transcripts in the different clusters using Velocyto with the raw data. We observed a high percentage of spliced RNA in the Germ cells B cluster compared to all the other clusters (Fig 4C, p = 1.35e-08, Chi-square contingency test). Only replicates 3 and 4, for which ATAC-seq data were available, are depicted in Fig 4C, but replicates 1 and 2 also show the same pattern (S12 Fig). The higher rate of spliced RNA can be linked to a decrease in or a complete pause of transcriptional activity, in line with the observed reduction in chromatin accessibility. While such a pattern has not been reported in similar datasets, running Velocyto on the Fly Cell Atlas ovary data also yielded a much higher percentage of spliced transcripts in the young germ cells cluster of *Drosophila* than in other cell types (S13 Fig). In order to explore the dynamics and time of onset of this putative transcriptional silencing, we performed pseudotime analysis using germ cells of replicates 3 and 4 (nuclei assigned as Escort cells in those replicates were included as they appear to be wrongly clustered Germ cells A and B)(Fig 4D). We see that both the ATAC and the RNA counts decrease across the germline pseudotime, suggesting that transcriptional repression is progressively established in the Germ cells A cluster (Fig 4E and 4F). The RNA counts seem to rise towards the end the germline pseudotime, which could be a sign of transient transcriptional reactivation, similar to what happens in *Drosophila* oocytes during oogenesis, between stages 9 and 11 (prophase I arrest ends at stage 13) [9]. However, as the ATAC counts do not show a similar pattern, the pattern may not be biological.

## 4. Downregulation of the Z chromosome in the germline

After identifying the different cell types in our dataset, we explored the expression of the differentiated region of the Z chromosome (A previously identified ~13 MB Z-linked region with half the coverage in females compared to males, which we refer to as S0 [52]) to assess whether meiotic sex chromosome inactivation is present in *Artemia*. Fig 5A shows the inferred S0/Autosomal values per nucleus on the UMAP, and highlights a strong excess of cells where the S0 seems to be downregulated in the two germ cell clusters. To explore this pattern quantitatively, we estimated the ratio of the mean expression of the genes in the S0 region (446 genes) and mean expression of Autosomal genes (26439 genes). The boxplot of the S0/Autosome expression shows that the Z-specific region is indeed downregulated in the germline cell

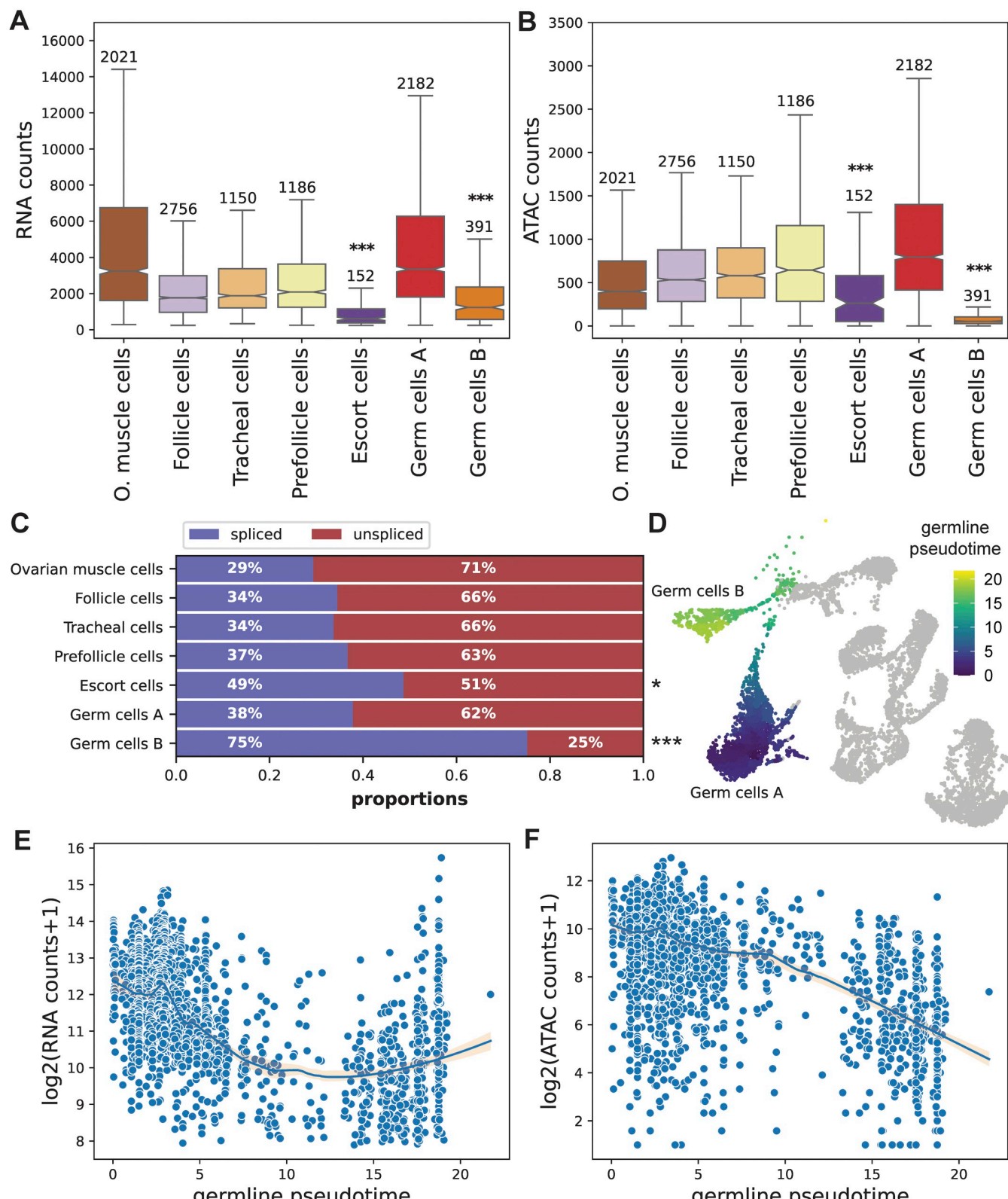

**Fig 4. Unique features of germ cells B.** A) The snRNA-seq counts per cell in the different clusters (replicates 3 and 4, number of nuclei above each boxplot). The stars show the significance values for group 3 and somatic clusters comparisons (Wilcoxon rank-sum test). B) The snATAC-seq counts (in peaks) in the

different clusters (replicates 3 and 4, number of nuclei above each boxplot). The stars show the significance values for group 3 and somatic clusters comparisons (Wilcoxon rank-sum test). C) The percentage of spliced and unspliced transcripts (replicates 3 and 4). The stars show the significance values for group 3 and somatic clusters comparisons (Chi-square contingency test). For the stars, *** denotes p-value < = 0.001, ** denotes p-value < = 0.01, and * denotes p-value < = 0.05. D) pseudotime analysis (Germ cells A, Germ cells B and escort cells) using replicates 3 and 4 E) RNA counts in Germ cells A, B, and escort cells across the germline pseudotime (the line depicts the local regression result with confidence intervals) F) ATAC counts in germ cells A, B, and escort cells across the germline pseudotime (the line depicts the local regression result with confidence intervals).

clusters compared to somatic cells (p<10–16 with Wilcoxon rank-sum test; Clusters belonging to groups 2 and 3 are used as the somatic control). We recover the same S0/Auto patterns when using the mean of non-overlapping bins of 446 autosomal genes instead of the mean of all autosomal genes (See methods section: "Estimation of S0/Autosomal ratio using non-overlapping autosomal windows" and S14 Fig).

The median of the expression ratio in Germ cells B is around 0.5, which is overall more consistent with lack of dosage compensation than with true Z inactivation. In order to check if any cells have expression patterns consistent with additional repression of the S0, we classified all cells in each cluster into bins of decreasing S0:A expression ratio that should reflect the presence of dosage compensation and/or repression of the S0 (S0/Auto ratio for all cells is adjusted by adding 1-median(S0/Auto of somatic clusters)):

- Complete or partial dosage compensation: S0/Auto > 0.66.

- Lack of dosage compensation: S0/Auto < = 0.66 and S0/Auto > 0.33.

- Repressed: S0/Auto < = 0.33.

As expected, the vast majority of cells of the somatic clusters have full dosage compensation, with virtually none being classified as Z-repressed. Germ cells A are enriched for lack of dosage compensation (27.83%, p = 0.0003 with Chi-square contingency test). Germ cells B show a high enrichment for lack of dosage compensation (55.24%, p = 1.07e-12), but we also observe a high proportion of cells with expression consistent with repression of the differentiated region (16.79%, p = 0.0002 compared with autosomal clusters). As an additional measure, we used percentile-based cutoffs to control for the heterogeneity of Z chromosome regulation status caused by noise, and we recovered the same enrichment patterns (See methods section "Z-chromosome regulation status using percentile-based cutoffs" and S15 Fig).

While these results point to a small subset of germline cells (less than 20% of the cells in the cluster) showing at least partial Z-inactivation, it is notoriously difficult to fully exclude that absence of dosage compensation, along with sparse and noisy data, could create such a pattern. To bypass this, we reasoned that absence of dosage compensation should only affect the S0 region of the Z, which no longer has W-homologs, whereas both younger non-recombining but undifferentiated regions (S1 and S2), as well as the pseudoautosomal region, should not be affected. On the other hand, depending on the mechanism at play, inactivation may spread to other regions of the chromosome. We therefore explored the expression of the other regions of the Z-chromosome, which include the pseudoautosomal region (PAR), and the younger strata S1 and S2 [52] (S16 Fig). Both germline clusters show lower expression levels of these undifferentiated regions compared to somatic clusters, with a more consistent downregulation in Germ cells B.

Finally, to explore whether the downregulation of the Z chromosome in germ cells corresponds a change in chromatin accessibility, we pooled the counts from each cluster together and counted the number of ATAC fragments in the differentiated and pseudoautosomal regions of the Z in windows of 500,000 bp and compared them to the number of fragments in autosomal windows (Fig 6A). We see a slight decrease in accessibility of the two germline

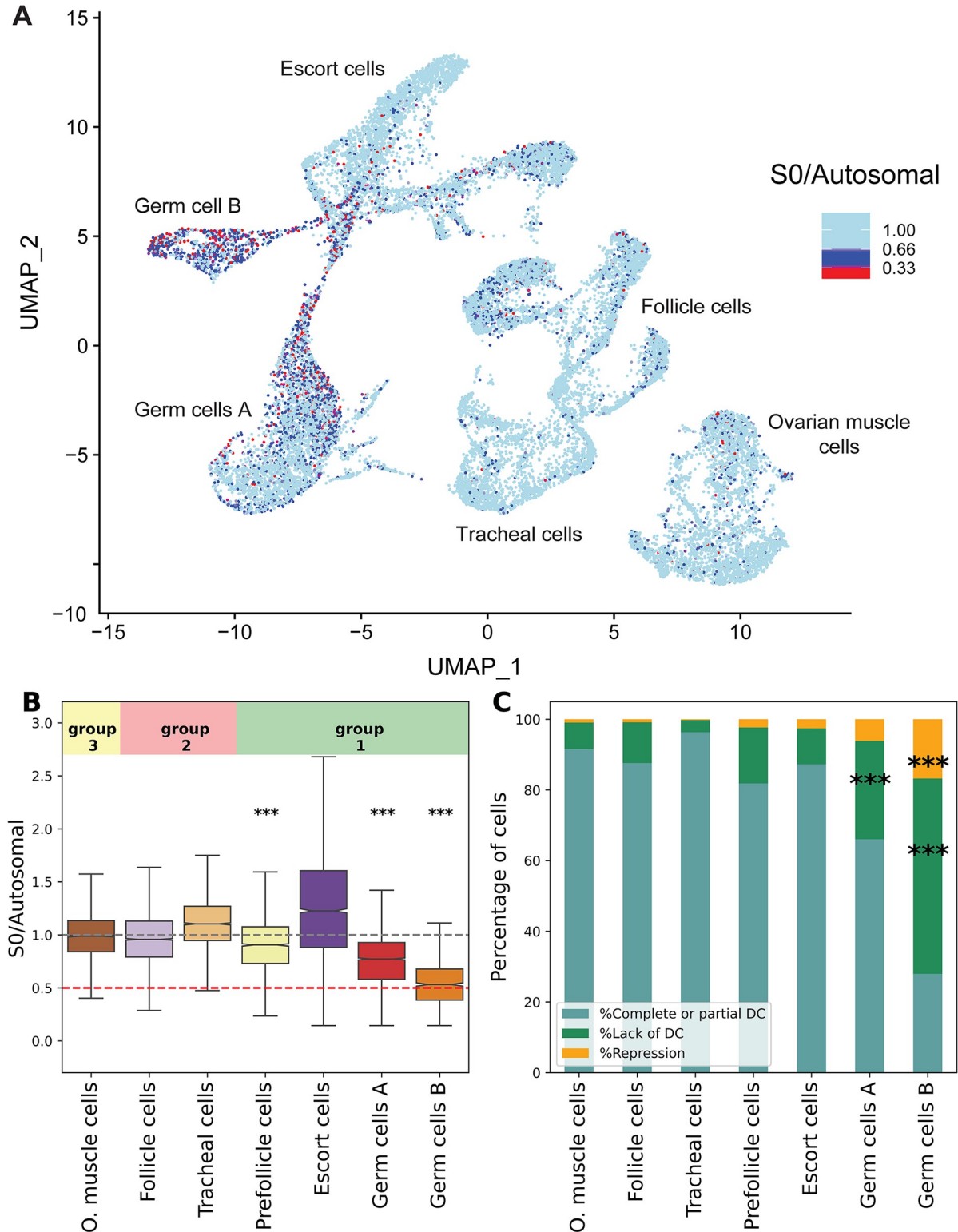

**Fig 5. Downregulation of the Z chromosome in the germ cells.** A) UMAP showing the log2(S0/Autosomes) expression per cell. B) The mean(S0)/mean(Autosomes) expression per cell estimated using the normalized counts matrix. The stars show the significance values for group 3 and somatic clusters comparisons (Wilcoxon rank-sum test). C) The percentage of cells that are dosage compensated (DC), lack dosage compensation (Lack of DC) and repressed (Repression). The stars show the significance values comparing group 1 clusters and somatic clusters (%Lack dosage compensation vs rest, and %Repression vs rest using Chi-square contingency test). For the stars, *** denotes p-value < = 0.001, ** denotes p-value < = 0.01, and * denotes p-value < = 0.05.

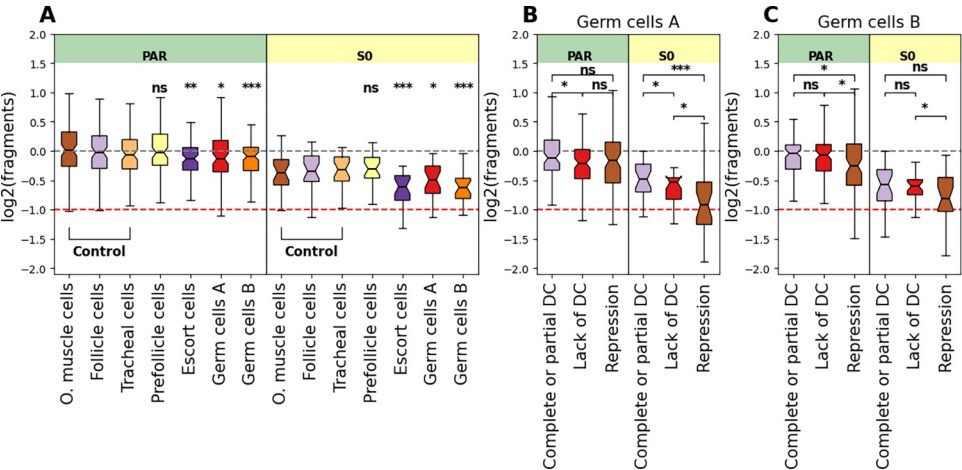

**Fig 6. Decreased Z-chromosome accessibility in the germline clusters.** A) log2(fragments) for autosomal, PAR, and S0 in windows of 500,000 bp estimated from pseudo-bulks of all nuclei within a cluster adjusted by subtracting the median(log2(autosomal windows)) of each cluster. The stars show the significance values for the comparisons between group 3 clusters and somatic clusters (Wilcoxon rank-sum test). B) and C) log2(fragments) for autosomal, PAR, and S0 in windows of 500,000 in Germ cells A and B split into pseudo-bulks based on the expression zone of the Z chromosome adjusted by subtracting the median(log2(autosomal windows)) of each cluster. The stars show the significance values for comparisons between the different categories (Wilcoxon rank-sum test). For the stars, *** denotes p-value < = 0.001, ** denotes p-value < = 0.01, and * denotes p-value < = 0.05. The red line is at -1, and corresponds to a two fold decrease in the number of fragments.

clusters compared to the somatic clusters in the S0 (p-value of 0.018 and 4.55e-06 for Germ cells A and B clusters respectively) and PAR (p-value of 0.037 and 0.00018 for Germ cells A and B clusters respectively) regions, which suggests that the Z chromosome is less accessible than the autosomes. As dosage compensated cells may be masking the signal when pooling all the counts together, we split the cells based on their Z expression zone (Fig 6B and 6C). Although most of the comparisons are not significant, we still see that the number of fragments is consistent with the expression zone, with the repressed cells having the lowest number of fragments originating from the S0 and PAR regions.

# Discussion

## Conserved cell identity programs across Arthropoda

In this study, we used single nucleus RNA sequencing to resolve the cellular complexity of the *Artemia* ovary. We identify clusters of cells with distinct expression patterns and show that they share different levels of expression-based homology with the *Drosophila* ovarian clusters from the Fly Cell Atlas. This suggests that many of the expression programs that give rise to cellular identity and function are highly conserved despite ~505 MYA (CI: 474.8–530.0 MYA) of divergence between Crustacea and Insecta [53]. Ovarian muscle cells show the highest level of conservation, in line with a previous study that showed a high similarity in orthologous gene expression of muscle cells across several vertebrate and invertebrate species [54]. Two clusters, which show some expression similarity to each other (S5 Fig), map to Tracheal and Follicle cells in *Drosophila* (epithelial cell types). In *Drosophila*, Tracheal cells form the tracheal system, which transports oxygen to the different organs [55], while follicle cells are involved in many aspects of oogenesis, including control of egg shape, eggshell formation, and formation of appendages, such as the dorsal appendage [56]. The two identified clusters (in addition to prefollicle cells) express tracheless (trh, S17A Fig), which is essential for the initiation and

maintenance of invagination [57]. The tracheales ortholog has also been found to be expressed in the salt gland of nauplii and in the thoracic epipod of juveniles in *Artemia franciscana*, suggesting it might play other roles in this species [58].

Early germ cells (germ cells A) also show a high level of conservation between the two species. Previous studies have linked the observed conservation of several aspects of germline cyst development between distant species, such as of mice and *Drosophila*, to the critical role of germ cells in the preservation of the nuclear genome and the importance of early oocytes for embryonic development [59]. We identify a single cluster in our dataset that maps to all the late stage germline clusters in *Drosophila*, including nurse cells. This is in line with expression patterns in this species, where all late germline and nurse cell clusters are highly correlated (S18 Fig). It should therefore be noted that our germline clusters may also contain a mix of developing oocytes and closely related nurse cells.

Some of the cluster annotations should be interpreted more cautiously, as they likely represent cell types with less conserved transcriptional programs, making inferences based on *Drosophila* less reliable. The *Artemia* cluster mapping to Escort cells is nested with the two germline clusters in the dendrogram, unlike Escort cells in the *Drosophila* dendrogram, which are nested with somatic cells (S5 Fig). The absence of Escort cells from the unmated females suggests those cells are either late stage follicle cells, late stage germline/embryonic cells, or sperm (the minority of cells that are assigned to the same cluster in replicates 3 and 4 after integration are likely to be misclustered germ cells A or B). We did not find any enrichment of testis biased genes in any of the different clusters, which rules out contamination by sperm (S19A and S19 Fig), but could not distinguish between other possibilities. The cluster mapping to the *Drosophila* prefollicle cells had the lowest alignment rate (0.230), and it also shows some expression similarity to the germ cell clusters (S5 Fig), suggesting the assignment as follicle cells may not be fully accurate. However, these cells have high expression of phagocytosis and apoptotic cell clearance genes *draper* and *draper-like* (S17D Fig), which are expressed in follicle cells to promote nurse cell death in *Drosophila* [60].

## The unusual regulation of the autosomes and Z chromosome in germline cells

We use 10x single cell RNA-seq and ATAC-seq to explore the expression and chromatin accessibility changes during female oogenesis in *Artemia* brine shrimp. We observe a dramatic decrease in the ATAC counts in the Germ cells B cluster, along with a noticeable decrease in the total RNA counts and the percentage of unspliced RNA. Those observations further support the idea that this cluster consists at least in part of late prophase cells, and the lack of ATAC signal is likely due to the compaction of chromatin and the establishment of prophase I arrest. A similar pattern has been observed in *Drosophila*, where the total number of ATAC peaks decreases dramatically in the later stages of oogenesis (whole ovaries) compared to GSCs and young ovaries [51]. In mouse single cell data, the number of ATAC peaks in mitotic cells decreases dramatically in the progression from prophase I to metaphase I [61]. The compact structure of the chromatin during mitotic and meiotic prophase is thought to present a barrier to many transcription factors, which causes a reduction in the levels of gene expression [9,62]. In meiosis, the global silencing of transcription in oocytes is highly conserved, and the extensive remodeling of the oocyte chromatin seems to play an important role in the oocyte to embryo transition [63]. We also checked the expression of three genes that have been shown to play an important role in the chromatin remodeling of the *Drosophila* oocyte (their knockdowns introduced significant disturbances to the oocyte epigenome) [9]. Lid, which is associated with the activating histone mark H3K4me3 is expressed in germ cells A, and Ash1 and

Bap1, associated with the repressive histone mark H3K27me3, are expressed in Germ cells B (S17E Fig).

In many species, gametogenesis coincides with the loss of dosage compensation. In the case of female mammals, this takes the form of reactivating the silenced X in the germline cells [18,19]. In *Drosophila* males, the lack of dosage compensation manifests in the absence of X chromosome upregulation in primordial germ cells, spermatocytes and spermatids [16,17,64]. This could be the result of the global reprogramming of the epigenome required for the generation of a "clean slate" for transmission to the embryo. How such a clean slate is achieved in the presence of ZW chromosomes is unclear, as loss of dosage compensation in oocytes could lead to imbalances in expression that are then transmitted to the embryo maternally [65]. In our data, the two germline clusters (germ cells A and germ cells B) show lower S0/Autosomal expression (Fig 5B), which seems to be driven by the enrichment in cells with S0 downregulation, consistent with a lack dosage compensation compared to somatic clusters. The ATAC-seq results show a similar pattern, where the germline cells have fewer counts in the S0 region compared to the somatic clusters (Fig 6). Other mechanisms must therefore be in place to avoid imbalances in the expression of maternal RNAs, such as their production by compensated nurse or follicle cells.

## Z chromosome repression in the germline

We find that the two germline clusters include cells which seem to have very low S0 expression (consistent with repression), and when we split the ATAC counts based on the expression zones, the repressed cells seem to have lower counts in the Z-specific region, consistent with the lowest accessibility. The fact that the whole Z-specific region seems to be downregulated and less accessible suggests that a whole chromosome mechanism may be in action, reminiscent of meiotic sex chromosome inactivation. Additionally, lack of dosage compensation in other species seems to result in less than 2 fold decrease in the expression of X/Z-linked genes (~1.5 in the *Drosophila* testes and ~1.6 in the chicken gonads) [64,66–68]. In our analysis, the distribution of S0/Autosome ratios per cell in the germ cells B cluster is centered at 0.5 (2-fold decrease in expression). If one assumes an expected value of 0.66 for lack of DC, then the observation of 0.5 might suggest a combination of lack of DC and sex chromosome repression. It is important to note some limitations of our data, including the low capture level of total mRNA per cell (high dropout rate), high ambient RNA, and sparse read mapping, which make confident inferences of silencing difficult. We used the same approach and percentile-based thresholds to check whether we see a similar pattern in the *Drosophila* testes dataset [69], and we only observe an enrichment in cells lacking dosage compensation in some of the germline clusters (mainly in the meiotic and post-meiotic cell types), but no cells show extreme repression of the X chromosome (S20 Fig). Additionally, we explored the expression of the *Artemia* genes annotated under the 'Facultative heterochromatin assembly' GO and 'Constitutive heterochromatin assembly' and the majority show Germ cells B specific expression pattern (S17B and S17C Fig). Taken together, our data therefore generally points towards the possibility that repression of the sex chromosome occurs during oogenesis, although a demonstration that repressive chromatin marks are present on the Z will be needed to confirm this.

The evolutionary hypotheses stemming for the mammalian case of MSCI sparked a lot of interest in understanding the conditions that favored the evolution of such a mechanism, and whether it is a universal consequence of having heteromorphic sex chromosomes. The fact that the reports of MSCI in *Drosophila* and chicken have been disputed later, and in many other species, such as moths and butterflies, the evidence so far suggests its absence, implies that the mechanism is either not as universal as initially assumed or that those species are

exceptions to the rule. In particular, both *Drosophila* males and *Lepidoptera* females have achiasmatic meiosis [70,71], and the chicken ZW chromosomes achieve complete heterologous synapsis [29]. *Artemia* have similar recombination rates in males and females, arguing against achiasmy in females, perhaps providing an explanation for why meiotic sex chromosome silencing may have been favored. More generally, broader sampling is needed to understand the role of the sex chromosome system (female or male heterogamety), the extent of sex chromosome differentiation (homomorphic or heteromorphic), the meiotic idiosyncrasies (type of pairing and presence or absence of recombination), and repeat content/meiotic driver presence/activity in promoting the evolution of MSCI. Our study of meiotic sex chromosome regulation in a female heterogametic system with a well differentiated region is a step in this direction. Our work also highlights single-nucleus RNA sequencing as a useful alternative to traditional approaches, such as epigenetic profiling and RNA-FISH, for identifying promising models for the study of meiotic sex chromosome regulation in species where it is difficult to isolate/identify nuclei of meiotic cells.

## Methods

### Single-nucleus sequencing of the *Artemia* Female reproductive system

We isolated *Artemia franciscana* adult females from either a colony or from vials where they were kept individually (see below), and washed them in Milli-Q water to remove any excess salt. The ovaries were dissected in ice-cold Dulbecco's phosphate-buffered saline (DPBS) and then moved to a 1.5 mL Eppendorf with DPBS and placed on ice. The sample was then washed once with DPBS, and after spinning down, the DPBS was removed without disrupting the pellet of ovaries, and 1 mL of the homogenization buffer was added to the sample. Following the protocol described in [72], the whole content of the Eppendorf was then transferred to a 1 mL Dounce homogenizer. The nuclei were then released by 20 strikes with the loose Dounce pestle and 40 strikes with the tight pestle on ice. The sample was then filtered through a 35 um cell strainer into a FACS tube, and then filtered again using a 40 μm Flowmi cell strainer into a 1.5 mL Eppendorf. Each sample was then centrifuged for 10 minutes at 4°C and 1000 g. The supernatant was discarded and the pellet was resuspended using ~300–500 μL of resuspension buffer. For the 10x Multiome samples, 10 μL/mL of 0.5% Digitonin (BN2006, Invitrogen) was added to the homogenization buffer to permeabilize the nuclei and facilitate the access of the tagmentation enzyme to the chromatin; the samples were incubated in the buffer for 5 minutes after homogenization before proceeding with the remaining steps. The samples were transferred for 10x genomics sorting and sequencing at the Vienna BioCenter Next Generation Sequencing (NGS) Core Facility. In all the replicates, 16,000 nuclei were loaded on the chip, targeting 10,000 individual nuclei.

For the 3' GEX experiments, 25 mated females were used in each replicate, and for the two replicates of the 10x Multiome ATAC+Gene expression experiments, the same number of unmated females (isolated at the Naupliar stage and maintained in individual vials until they reached sexual maturity) were used per replicate. As our experiments include mixed genotypes, it was possible to estimate the percentage of ambient RNA in each replicate using Souporcell [73](S2 Table).

### Preprocessing, Quality Control, and Integration of the different replicates

The reads from each sample were mapped to the *A. franciscana* genome [52], annotated using StringTie2 [74], using 10x Genomics Cell Ranger 5.0.0 for the two 3'GEX samples and using Cell Ranger ARC 2.0.2 for the two Single Cell Multiome ATAC + Gene Expression samples [75,76]. The CellBender v0.2 [77] package was run on the raw gene-by-cell matrix from each

replicate to remove the technical artifacts and background noise and produce an improved estimate of gene expression per cell. Specific low count thresholds, droplet training fractions, and false positive rates were chosen for each sample following the CellBender best practices (https://cellbender.readthedocs.io/en/latest/troubleshooting/), and are provided in the GitHub page. The output of CellBender was then loaded into Seurat [41], where nuclei with < 10 features, nuclei with > 3% mitochondrial content, and doublets were removed. The filtered nuclei from all the replicates were then loaded into Seurat, and only nuclei with (nFeature_RNA > 200 & nFeature_RNA < 25000) were retained. The highly variable features were identified using DUBStepR [78] with default parameters and the replicates were integrated using Harmony [79], clustered using graph-based approaches, and then visualized using non-linear dimensionality reduction UMAP. The resolution for clustering (0.05) was determined using the marker specificity-based analysis from scMiko [40]. The cluster markers were identified using two different Seurat functions: FindConservedMarkers and FindAllMarkers. As FindConservedMarkers identifies the differentially expressed genes between the clusters which are conserved across the replicates, we reasoned that the results would not be reliable in the case of Escort cells due to their absence from replicates 3 and 4. Therefore, we also used FindAllMarkers, which does not take the replicate information into account. To ensure that our results are not an artifact of ambient RNA removal, we performed the analysis with the raw counts. The global structure is preserved (S21 Fig), along with the significant differences between the identified dosage compensated, not dosage compensated, and repressed nuclei (S22 Fig). Additionally, despite the noisiness of the raw data, the enrichment in Orb and Vas was clear in the germline cells compared to the somatic clusters (S21B and S21C Fig). The single-nucleus gene expression atlas and metadata can be viewed on the UCSC Cell Browser [80].

## ATAC-seq clustering and Analysis

For the clustering analysis, the raw count matrix was loaded into Seurat and filtered to keep only the cells that are in the expression clusters. The peaks were called per cluster using MACS2 [44] in each replicate separately and the resulting peaks were then combined. The data was then normalized using RunTFIDF (method = 3), and the variable features were identified using FindTopFeatures with min.cutoff = 'q3'. RunSVD was then used to perform latent semantic indexing (LSI) and the nonlinear dimensionality reduction was performed using UMAP. The same clustering as for the gene expression analysis was used for visualization. We have also checked the correlation between the peaks and expression. We divided the genes in each cluster into three categories ($< = 20^{th}$ percentile for low expression genes, $>20^{th}$ and $<80^{th}$ percentiles for medium expression, and high expression genes $> = 80^{th}$ percentile). We used the mean of the ATAC counts in the linked peaks for each gene within a cluster, and S23 Fig shows that the peak enrichment corresponds to the expression level in all the clusters (Germ cells B and Escort cells do not show as clean a pattern due to the low number of peaks detected).

## Integration of the *Artemia* atlas with the *Drosophila* Fly Cell Atlas

We used the SAMap blast-based mapping script to map the *Artemia* transcripts to the *Drosophila* CDS (dmel-all-CDS-r6.31.fasta) downloaded from FlyBase [81] and filtered to keep only the longest isoform for each gene. The *Drosophila* single nucleus data (10x VSN Ovary (Stringent), 10x genomics, H5AD) was downloaded from: https://cloud.flycellatlas.org/index.php/s/zgZe3Zsegpn5Bpg/download/s_fca_biohub_ovary_10x.h5ad. SAMap [43] was then run using the Jupyter notebook provided on the GitHub page. An alignment threshold of 0.2 was used for displaying the cluster correspondence using the Sankey diagram.

### Identification of meiosis and germline markers

The germline and meiosis markers were found in the literature [48,47] and the *Artemia* homologs were identified as the reciprocal best hits based on the SAMap mapping output. The expression of the markers in *Drosophila* is shown in S24 Fig.

### Networks analysis using hdWGCNA

In order to construct the co-expression network, we ran hdWGCNA [50] on genes expressed in at least 5% of the nuclei. We constructed the metacells grouping by the cell type and replicate information, and we constructed the co-expression network for all the clusters simultaneously. We performed differential module eigengene (DME) analysis comparing the germ cells A and germ cells B group to a group made of all the other clusters. We then performed pseudotime trajectory analysis on the whole dataset, isolated the germline cells (Figs 3B and S25) and explored the module dynamics across the pseudotime (Figs 3C and 3D and S26).

### Quantifying the proportion of spliced and unspliced transcripts

We used Velocyto [82] to annotate spliced and unspliced transcripts and generate spliced/unspliced count matrices for each replicate using the output from Cell Ranger. We then used SCANPY [83] to merge the matrices, and scVelo [84] to plot the proportions of spliced/unspliced counts (Jupyter notebook provided on the GitHub page). For the *Drosophila* estimation, we downloaded the raw ovary Fastq files (NCBI BioProject PRJEB45570), aligned them to the *Drosophila* genome (Drosophila_melanogaster.BDGP6.32.dna.toplevel.fa) using 10x Genomics Cell Ranger 5.0.0, and used Velocyto to get the spliced and unspliced counts, and then merged the matrices with the expression matrix provided on the Fly Cell Atlas (scripts provided on GitHub page).

### Protein Interaction network and GO enrichment analysis

We translated the *Artemia franciscana* transcriptome generated using StringTie2 with the Perl script GetLongestAA_v1_July2020.pl, and the translated sequences were uploaded to https://string-db.org/, where the PPI and GO enrichment analyses for the modules were performed. The annotated proteome is accessible using the following link: https://version-12-0.string-db.org/organism/STRG0A95DBT.

### Z-chromosome regulation status using percentile-based cutoffs

As the within cluster variation in the status of Z-chromosome expression is possibly driven by noise, we implemented more conservative thresholds that apply a 5% false positive rate to the first category in each comparison to provide a noise-sensitive estimate of the cluster-specific enrichments (S15 Fig):

- Complete or Partial dosage compensation: S0/Auto > 5th percentile of S0/Auto in somatic clusters (>0.57).

- Lack of dosage compensation: S0/Auto < = 5th percentile of S0/Auto in somatic clusters and S0/Auto > (5th percentile of S0/Auto in somatic clusters)/2 (< = 0.57 and >0.28).

- Repressed: S0/Auto < = (5th percentile of S0/Auto in somatic clusters)/2 (< = 0.28).

## Estimation of S0/Autosomal ratio using non-overlapping autosomal windows

To ensure that our S0/Autosomal expression estimates are not affected by the low expression throughout the genome, as is the case for some germline cells (see results section 3), we divided the genome into 49 non-overlapping sliding windows with the same number of genes as the S0 (446 genes). We reasoned that regions of similar gene counts as the S0 are as susceptible to the low detection rates and non-biological zeros that affect single-cell RNA-seq data, and can therefore be used to ensure the overall patterns are not technical artifacts. In S14 Fig, we show the distribution of the per cluster medians of S0/Autosomal window for all the 49 windows.

## Supporting information

**S1 Fig. UMAP of all nuclei colored by replicate.**
(TIF)

**S2 Fig. The results of the specificity-based resolution selection analysis.** A) specificity scores. B) Specificity curves. C) Dot plot of top cluster-specific markers.
(TIFF)

**S3 Fig. Cluster specific markers based on findallmarkers.**
(TIFF)

**S4 Fig. Cluster specific markers based on findconservedmarkers.**
(TIFF)

**S5 Fig. A dendrogram and heatmap of the correlation matrix of the mean expression values per cluster (*Artemia*).**
(TIFF)

**S6 Fig. The RNA-seq and ATAC-seq UMAPs for replicates 3 and 4.** A) UMAP of replicates 3 and 4 nuclei based on expression. B) UMAP of the nuclei from replicates 3 and 4 nuclei based on peaks.
(TIFF)

**S7 Fig. Dot plot depicting the expression of the modules identified using the co-expression network analysis.**
(TIF)

**S8 Fig. Biological process GO enrichment in module 6 (433 genes).**
(TIFF)

**S9 Fig. Biological process GO enrichment in module 2 (1363 genes).**
(TIFF)

**S10 Fig. Biological process GO enrichment in module 12 (61 genes).**
(TIFF)

**S11 Fig. The number of ATAC fragments in each cluster for replicates 3 and 4.** A) ATAC fragments per cell for all clusters in Replicate 3. B) ATAC fragments per cell for all clusters in Replicate 4.
(TIFF)

**S12 Fig. The percentage of spliced and unspliced transcripts (replicates 1 and 2).**
(TIFF)

**S13 Fig. *Drosophila* estimates of spliced and unspliced transcripts in the Fly Cell Atlas data (raw data downloaded from NCBI BioProject PRJEB45570).**
(TIFF)

**S14 Fig. Cluster medians of mean(S0)/mean(window) per cell) estimated using 49 non-overlapping genomic windows with the same number of genes as the S0.**
(TIFF)

**S15 Fig. The percentage of cells that are dosage compensated (DC), lack dosage compensation (Lack of DC) and repressed (Repression) using percentile-based cutoffs.** The stars show the significance values comparing group 1 clusters and somatic clusters (%Lack dosage compensation vs rest, and %Repression vs rest using Chi-square contingency test). For the stars, *** denotes p-value < = 0.001, ** denotes p-value < = 0.01, and * denotes p-value < = 0.05.
(TIFF)

**S16 Fig. The expression patterns of the different regions of the Z-chromosome (PAR, S1, and S2) and W genes compared to autosomal expression.** A) The structure of the Z chromosome as described in (Bett et al., 2024), with the large pseudoautosomal region (PAR), the differentiated region (S0), and two younger strata (S1 and S2). B) PAR/Autosomes expression per cell C) S1/Autosomes expression per cell D) S2/Autosomes expression per cell E) W/Autosomes expression per cell. The normalized counts matrix was used for all the estimates.
(TIFF)

**S17 Fig. Dot plots depicting the expression of various relevant genes in the different clusters.** A) Trachealess expression dot plot. B) Facultative heterochromatin assembly network expression. C) Constitutive heterochromatin assembly network expression D) phagocytosis genes expression. E) Genes involved in the modeling of oocyte chromatin.
(TIFF)

**S18 Fig. A dendrogram and heatmap of the correlation matrix of the mean expression values per cluster in the Fly Cell Atlas ovary data (10x, Stringent, H5AD, downloaded from https://flycellatlas.org/).**
(TIFF)

**S19 Fig. Somatic clusters are enriched for genes that are female biased in heads (a somatic tissue), while germ cells B are enriched for female biased genes in the ovary.** Male and female-biased genes were inferred by running DEseq2 with standard parameters on the bulk RNA-seq data from (Huylmans et al., 2019). A) Differentially expressed male vs female heads (Filtered for >5,-5 fold change and <0.01 qval). B) Differentially expressed ovaries vs testes (Filtered for >10,-10 fold change and <0.01 qval).
(TIFF)

**S20 Fig. Downregulation of the X chromosome in the testis from the *Drosophila* single nucleus atlas.** A) The X/Autosomes expression per cell estimated using the normalized counts matrix. B) The percentage of cells that have partial or complete dosage compensation (Complete or partial DC), lack dosage compensation (Lack of DC), or repression using the percentile-based cutoffs. The testis snRNA-seq data (Raz AA et al., 2023) was obtained from: https://datadryad.org/stash/dataset/doi:10.5061/dryad.m63xsj454.
(TIF)

**S21 Fig. Analysis of the snRNA-seq data without ambient RNA removal.** A) UMAP with no ambient RNA removal. B) Orb expression no ambient RNA removal. C) Vasa expression with

no ambient RNA removal. The cells were labeled based on the annotation from the main analysis.
(TIFF)

**S22 Fig. S0/Autosome ratios without ambient RNA removal.** A) S0/Autosomes expression per cell estimated using the normalized counts matrix (No ambient RNA removal) B) S0/Autosomal for the germ cells A cluster cells in the different expression zones (no ambient removal). C) S0/Autosomal for the germ cells B cluster cells in the different expression zones (no ambient removal).
(TIFF)

**S23 Fig. Correlation between the mean expression of genes in each cluster and the mean of the ATAC counts in the linked peaks.** Genes in each cluster were split based on their expression into three categories: $< = 20^{th}$ percentile for low expression genes, $>20^{th}$ and $<80^{th}$% percentiles for medium expression, and high expression genes $> = 80^{th}$ percentile.
(TIFF)

**S24 Fig. The expression of vas and orb, and the genes involved in the different stages of meiosis in *Drosophila*, along with the expression of genes involved in protein production.** The plot was produced with Scanpy using the Fly Cell Atlas ovary dataset (10x, Stringent, H5AD, downloaded from https://flycellatlas.org/).
(TIFF)

**S25 Fig. Pseudotime trajectory analysis.** A) UMAP depicting the Monocle3 pseudotime trajectory of all the clusters (all replicates). B) germline pseudotime (Germ cells A and Germ cells B). Y_2 (panel A) was used as the principal node.
(TIFF)

**S26 Fig. Module expression dynamics across the germline pseudotime.**
(TIFF)

**S1 Table. Number of cells in the Expression UMAP from the females allowed to mate and the females not allowed to mate.**
(XLSX)

**S2 Table. Ambient RNA percentage per replicate estimated using Souporcell.**
(XLSX)

## Acknowledgments

We thank the Vicoso group for their valuable comments on the earlier draft of the manuscript. We would also like to thank the Vienna BioCenter Next Generation Sequencing (NGS) facility staff, and in particular, Thomas Grentzinger for his support with the handling and sequencing of the samples, the scientific computing unit at ISTA for the computational resources, Brittney Wick for the help with hosting our data on the UCSC Cell Browser, and Lora B. Sweeney for her valuable input at the different stages of the project.

## Author Contributions

**Conceptualization:** Marwan Elkrewi, Beatriz Vicoso.

**Data curation:** Marwan Elkrewi.

**Formal analysis:** Marwan Elkrewi.

**Funding acquisition:** Beatriz Vicoso.

**Investigation:** Marwan Elkrewi, Beatriz Vicoso.

**Methodology:** Marwan Elkrewi, Beatriz Vicoso.

**Project administration:** Beatriz Vicoso.

**Resources:** Beatriz Vicoso.

**Software:** Marwan Elkrewi.

**Supervision:** Beatriz Vicoso.

**Validation:** Marwan Elkrewi.

**Visualization:** Marwan Elkrewi.

**Writing – original draft:** Marwan Elkrewi, Beatriz Vicoso.

**Writing – review & editing:** Marwan Elkrewi, Beatriz Vicoso.

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
