## [Decision Letter · Decision Letter 0]

21 May 2024

Dear Dr Elkrewi,

Thank you very much for submitting your Research Article entitled 'Single-nucleus atlas of the Artemia female reproductive system suggests germline repression of the Z chromosome' to PLOS Genetics.

The manuscript was fully evaluated at the editorial level and by independent peer reviewers. Three experts have provided their review comments. While the reviewers agree the work offers novel insights into MSCI, they have raised several technical concerns that, once addressed, should strengthen the manuscript. Based on the reviews, we will not be able to accept this version of the manuscript, but we would be willing to review a much-revised version. We cannot, of course, promise publication at that time.

If you decide to revise the manuscript for further consideration at PLOS Genetics, please aim to resubmit within the next 60 days, unless it will take extra time to address the concerns of the reviewers, in which case we would appreciate an expected resubmission date by email to plosgenetics@plos.org.

We are sorry that we cannot be more positive about your manuscript at this stage. Please do not hesitate to contact us if you have any concerns or questions.

Yours sincerely,

Li Zhao

Guest Editor

PLOS Genetics

Kelly Dyer

Section Editor

PLOS Genetics

Reviewer's Responses to Questions

**Comments to the Authors:**

Reviewer #1: The authors performed single-cell RNA-seq and ATAC-seq in ovaries of the brine shrimp Artemia franciscana, a species with ZW sex chromosomes. Using the well-annotated cell atlas of Drosophila melanogaster, they show that reproductive cell types are well conserved across arthropods. Although the Z chromosome appears to be dosage compensated in somatic cells, where it has approximately equal expression as the autosomes, its expression is downregulated in germline cells. The reduced Z expression is consistent with a lack of dosage compensation in the germline and may also reflect sex-chromosome repression analogous to the MSCI that occurs in mammals.

Overall, I find this to be a well-written manuscript with clear results. It adds to our knowledge of reproductive system organization and sex chromosome gene expression using state-of-the-art methods in a non-model arthropod. Thus, it is a valuable contribution to the field, which consists mainly of studies in Drosophila and, to a lesser extent, Lepidoptera and Daphnia. The only part of the manuscript that I found difficult to understand was the section "Downregulation of the Z chromosome in the germline", which I will comment on below.

1. I am confused by the classification scheme presented in lines 295-300. After going over the 3 bullet points multiple times, I think there are several mistakes:

a) The first bullet point uses "log2(S0/Auto)", while the others use "S0/Auto". I assume it should be log2 in all cases, since some of the values given are negative.

b) In bullet point 2, the "and" statement doesn't make sense, because both percentile cutoffs are based on S0/Auto. Should the second part be "somatic/2".

c) Also in bullet point 2, should the first value be "<= -0.91", which would match the value in the first bullet point?

2. If I now understand the classification scheme as corrected above, it is rather conservative in classifying "Lack of DC" relative to "Complete DC", and "Repressed" relative to "Lack of DC". This is because the authors use an approach that essentially applies a 5% false positive rate to the first category in each comparison. I understand that this is a conservative approach and should not have a large effect on the statistical analysis that compares proportions between cell types with a chi-square test (although might reduce power in cases where there are a small number of cells in a category). However, it may give misleading results in terms of the percentage of cells in each category as given in lines 303-307. I don't think the authors need to change the way the results are presented in the text/figures, but they should acknowledge in the text that their approach will underestimate the percentage of genes in the "Lack of DC" and "Repressed" categories. Maybe they could add a supplemental figure similar to figure 5, but using more relaxed (albeit arbitrary) definitions that don't rely on outliers to an empirical distribution. For example (on a non-log2 scale),

Complete DC: S0/Auto > 0.66

Lack of DC: S0/Auto < 0.66 and S0/Auto > 0.33

Repressed: S0/Auto < 0.33

3. The above classification is also confusing, because the authors consider a log2(S0/Auto) > -0.91 as "complete" dosage compensation. However, at the low end this means the X has only slightly higher than half the expression of the autosomes. It might be better to consider this "Complete or partial DC".

Minor points

line 20: "Melanogaster" should not be capitalized

line 87: I don't think "Meiotic" needs to be capitalized here. Similarly, "Meiosis" does not need to be capitalized in line 181.

line 198: here "drosophila" should be capitalized and in italics

line 289: "The median log2 of the expression ratio in Germ cells B is around -1, which is overall more consistent with lack of dosage compensation than with true Z inactivation". This observation does match the expectation based on gene dose, however, previous studies in Drosophila and birds have observed that the ratio is not as extreme in the absence of DC. It appears that the X (or Z)/Auto is closer to -0.6 (e.g. Itoh et al. 2007; Ellegren et al. 2007; Meiklejohn et al. 2011; Argyridou and Parsch 2018). This would be expected if some genes had gene-specific regulatory mechanisms that balanced their expression. Many genes are part of regulatory networks that have feedback loops to maintain appropriate expression levels. Thus, I find it a bit surprising that Artemia seems to lack this type of regulation and expression appears to be almost entirely controlled by gene dose? If one assumes an expected value of -0.6 for lack of DC, then the observation of -1 might suggest a combination of lack of DC and sex chromosome repression.

Reviewer #2: Overview:

Elkrewi and Vicoso generated the snRNA-seq and snATAC-seq for Artemia ovaries. Uniquely, Artemia has the ZW sex chromosome system which provides the opportunity to determine whether there is MSCI of the Z. Despite little being known about the regulatory programs of Artemia germline, the authors used a myriad of cutting edge SC methods to establish cellt ypes based on similarities to the Drosophila ovaries, including two germ cell populations (A and B). While the expression genes in the S0 region (differentiated and presumably non-recombining)of the Z decreases (relative to autosomes) in the germ cell populations, the reduction is within the range of loss of dosage compensation instead of MSCI. Interestingly, the authors found a reduced accessibility of the PAR, despite DC being unnecessary, suggesting a chromosome-wide silencing/downregulation mechanism consistent with MSCI.

Comments:

As the authors described, the status of MSCI has been highly contested in several organisms (flies and chicken), despite being clear in others (C. elegans and eutherian mammals). The ZW sex chromosome system of Artemia is a great system to examine this enigmatic germline regulation. The absence of well understood expression of ovarian genes makes cell cluster identity classification particularly challenging. But the authors used several sophisticated approaches to map the cell clusters to the melanogaster ovarian atlas, revealing that despite the great evolutionary distance there is enough similarity in gene expression programs of cell types to enable cell type assignment. I really appreciate the lengths to which the authors took to ensure proper inference of cell types.

My primary concerns pertain to how X:A ratio is inferred and how to interpret reduced expression in relation to the dosage compensation status and how accessibility on the Z is interpreted, especially across stratas.

Primary concerns:

I have a hard time understanding how the authors estimated the Z:A ratio using autosomal “bins”. For each cell they first averaged across the 446 genes in the S0 region. But I do not understand how the denominator (autosomal gene expression) is generated in “bins with the same number of genes across the whole genome”. Since comparisons of Z:A ratio is a key part of the study, the authors must provide a better explanation of their procedure. In addition, what would the distributions look like with a simple ratio of S0 median to autosomal median?

The authors divided the germline cells into those showing complete dosage compensation (of S0), lack of dosage compensation, and repressed. Cells are considered to have “complete dosage compensation” if their log2(S0/Auto) is >-0.91. This equates to a lower-bound ratio of 0.532 which is typically considered no dosage compensation. The lack of dosage compensation is categorized by the cutoff of “<=0.91 and >-1.91” which equates to upperbound and lowerbound of 1.879 and 0.266, respectively which is nonsensical (Did the authors mean <= -0.91). The repressed category has a cutoff of <= -1.91 equating to a lower-bound ratio 0.266. I am not sure if there are typos, but these expression range classifiers are strange. There are also instances of switching from log scale to linear scale. E.g. “S0/Auto in somatic clusters -1”, do they mean log(S0/Auto) - 1?

Odd cutoffs aside, classification of cell types like this implies that cells within the same cluster can have different Z chromosome regulation status. While this could be the case, alternatively, the cell-by-cell variation in Z:A ratio could just reflect a noisy distribution (especially when overall readcount/expression is low as is the case for germ cells B) and the median/mean is closer to the true state of the cell population. A diagnostic way to show that there are indeed different population of cells (some with DC, some no DC, and some repressed/inactivated) in germ cells B would be a clear multi-modal distribution. A unimodal distribution would suggest that noise is driving the spread and cell type classification. In addition, it is also worth noting, if the Z-repressed cells indeed represent a distinct group of cells, the number of cells undergoing MSCI (or more accurately MSCRepression) appear to be a very small population of the germ cells.

The authors wisely looked at the PAR and younger stratas on the Y to infer MSCI. Loss of dosage compensation and MSCI should manifest in different patterns. The significant decrease of PAR expression is perhaps the most convincing evidence of repression to me, since it doesn’t need dosage compensation. As such, I would recommend the authors display Figure S12 as a main figure (and perhaps a schematic breakdown of the Z chromosome architecture based on their earlier study). However, there are other oddities that make it hard to interpret the PAR expression, including significantly reduced expression across many somatic cells, with prefolicle being significant. What would explain the unexpected downregulation in a somatic cell type? How many genes are on the PAR?

Using sn-ATACseq the authors argue that the S0 is less accessible as support for repression instead of loss of DC. While the reduced accessibility of the S0 is significant and notable, it can also be explained by the copy number difference, i.e. the PAR has two copies while the S0 has one.

Other concerns/questions/comments:

In the Drosophila testes, Y-linked genes are upregulated in the meiotic stages. This is particularly striking on a recently formed neo-Y chromosome and is thought to result from the masculinization of the male-specific chromosome (perhaps due to sexual conflict). In other testes cell types Y-linked genes are typically lowly expressed. However, the authors observed downregulation of the W in the meiotic stages instead and “dosage-compensated” expression in several of the somatic tissues. This is very counter-intuitive. How many genes are found on the W, and do the authors have an explanation for why they appear dosage compensated and become down regulated?

At several points in the manuscript, the authors state/imply there is no dosage compensation in the Drosophila male germline and cite several papers such as Witt et al 2021. Recent genomic and scRNA-seq studies have pretty clearly demonstrated the presence of DC during at least the early stages of the male germline, including Witt et al 2021, Mahadevaraju 2021, Anderson et al 2023, Wei et al 2024, and others.

In Line 74-75. The description of dosage compensation in the female germline of mammals should be better described than merely “disappear/reverse”. It is “reconfigured” and rather complex, including a period of hyper-transcription and then dosage balance via expression of both Xs.

The authors should consider discussing the results from a recent publication investigating germline dosage compensation loss vs MSCI in a Drosophila species with neo-sex chromosomes.

The cell grouping (1,2, and 3) is flipped in Figure 5B.

Line 259-260, the authors state “RNA counts decrease across the germline pseudotime”. However Figure 4E shows an increase in expression at the end of the pseudotime.

In Figure 6 What are red dotted lines supposed to represent?

Reviewer #3: In this manuscript, the authors performed single nucleus RNA-seq and ATAC-seq of the Artemia female reproductive system. This study presented a comprehensive single nucleus atlas of the adult Artemia franciscana female reproductive system. For example, the study identified different cell clusters, uncovered the gene expression dynamics across different cells, and shed light on the status of MSCI in the ZW system. Together with flycellatlas data, the authors also showed the conservation and divergence of female reproductive system between Artemia and Drosophila. However, I have some concerns before I would recommend the manuscript to be published.

1. There are a decent number of cells in the identified Germ cells B that can map to Artefact cells in flycellatlas data. Including these cells might be problematic, as these cells, according to the flycellatlas paper, expressed nearly all genes. The authors should clarify if the Artefact cells were included in subsequent analysis.

2. In Figure 4A and 4B, the authors concluded that transcription is repressed in late germ cells by comparing the number of RNA/ATAC counts in different cell clusters. I wonder if the authors used normalized RNA counts for each gene per cell. I assume that the authors used total RNA counts per cell cluster. This may raise concerns, as germ cells B cluster has much fewer number of cells and the comparison is not valid. For example the conclusion in line 399-405 would not stand if the authors used the total counts instead of normalized counts.

3. One of the major significances of the work is the discussion of the conservation and divergence between Artemia and Drosophila female reproductive system. In the introduction, the authors described and referred to literatures about the absence of MSCI in the male germline of fruit flies. In the results, the authors showed a clear downregulation of Z chromosome in the germline and repression of transcription in late germ cells. I wonder how much conservation and divergence in this pattern would be expected in Drosophila female germline cells.

4. In Figure 6, I am pretty sure the authors used autosomal windows to normalize the data. The authors should adjust the captions or labels in Figure 6 to reflect their analysis.

5. Line 89, the authors described that MSCI in chickens and fruit flies were disputed afterward. However, the authors did not list references for the absence of MSCI in fruit flies in fruit flies.

6. In line 277, the authors the expression of the differentiated region of the Z chromosome, to test whether MSCI presents in Artemia. The authors should briefly describe the term “differentiated region” in the main text, as this definition is important for readers to understand the whole idea.

7. Line 284, the authors stated that “we estimated the ratio of the mean expression of the genes in the S0 region (446 genes) and of bins with the same number of genes across the whole genome in each cell (Figure 5B shows mean(S0/bin1 +S0/bin2 +..+ S0/binN))”. This is unclear, as the axis label of Figure 5B is log2(S0/Autosomal). Can the authors elaborate this?

8. Related to the point above, the authors should make clear descriptions as for the metrics used and how they were used in Figures 4-6. The descriptions in the current main text were confusing.

9. Line 412, the authors checked three genes that play important roles in chromatin remodeling of the Drosophila oocyte. The authors should cite the original work(s). I am also curious if there are other important genes and if these genes are expressed in Drosophila.

10. In Figure 3D, can the authors elaborate “Module Eigengene”? In Figure 3E and 3F, the authors used strength of enrichment, could the authors elaborate how they performed Go analysis and how the strength was calculated? I would assume that the PPI enrichment p-value were obtained from the STRING database, can the authors clarify?

11. Related to the point above, in line 219, the authors stated that “3 had significant enrichment”. Does this mean significant PPI enrichment or “Go analysis” enrichment? As for the 3 modules, I guess they are module 6, 7, and 9 since the authors showed the enriched biological processes of the three modules. Other than the three modules, module 2 (germ cells A specific) and 12 (germ cells B specific) seem to be significant too. The authors should clarify why the two modules are ignored in further analysis.

12. Line 297, did the authors mean “<=-0.91” instead of “<=0.91”? Can the author also elaborate somatic cluster -1?

13. Figure 6 was not mentioned in the text. Did the author mean Figure 6 instead of Figure 5 in line 344 and line 349.

14. Line 425-427, the authors should refer to Figure 6 here.

15. Line 542, the authors should cite the literatures.

**Have all data underlying the figures and results presented in the manuscript been provided?**

Reviewer #1: Yes

Reviewer #2: None

Reviewer #3: Yes

PLOS authors have the option to publish the peer review history of their article (what does this mean?). If published, this will include your full peer review and any attached files.

Reviewer #1: No

Reviewer #2: No

Reviewer #3: No

---

## [Decision Letter · Decision Letter 1]

26 Jul 2024

Dear Dr Elkrewi,

We are pleased to inform you that your manuscript entitled "Single-nucleus atlas of the Artemia female reproductive system suggests germline repression of the Z chromosome" has been editorially accepted for publication in PLOS Genetics. Congratulations!

Yours sincerely,

Li Zhao

Guest Editor

PLOS Genetics

Kelly Dyer

Section Editor

PLOS Genetics

Comments from the reviewers (if applicable):

Three experts found your revision adequate and satisfactory. I thus am happy to recommend publication. Please make sure all relevant data files and codes are public before official publication. Congratulations, Marwan and Beatriz.

Reviewer's Responses to Questions

**Comments to the Authors:**

Reviewer #1: The authors have done a good job of addressing my comments on the previous version. Their revisions have improved the manuscript, which was already of high quality. I have no further comments.

Reviewer #2: The current version of the manuscript is much improved and I thank the authors for their care and thoroughness with the revised analyses and clarifications.

It is up to the author but, I am still of the opinion that the PAR expression deserves to be included in main text, since it is aparticularly diagnostic contrast for the change in regulatory regime of the W across the germline

Reviewer #3: The authors have addressed my concerns adequately. Congratulations to the authors for their work and thank you to the authors for their contributions.

**Have all data underlying the figures and results presented in the manuscript been provided?**

Reviewer #1: Yes

Reviewer #2: None

Reviewer #3: Yes

PLOS authors have the option to publish the peer review history of their article (what does this mean?). If published, this will include your full peer review and any attached files.

Reviewer #1: No

Reviewer #2: No

Reviewer #3: No

**Data Deposition**

http://datadryad.org/submit?journalID=pgenetics&manu=PGENETICS-D-24-00341R1

**Press Queries**

---

## [Editor Report · Acceptance letter]

26 Aug 2024

PGENETICS-D-24-00341R1 

Single-nucleus atlas of the Artemia female reproductive system suggests germline repression of the Z chromosome 

Dear Dr Elkrewi, 

We are pleased to inform you that your manuscript entitled "Single-nucleus atlas of the Artemia female reproductive system suggests germline repression of the Z chromosome" has been formally accepted for publication in PLOS Genetics! Your manuscript is now with our production department and you will be notified of the publication date in due course.

With kind regards,

Zsofia Freund

PLOS Genetics

On behalf of:
